# Unequal cluster sizes in stepped-wedge cluster randomised trials: a systematic review

Caroline Kristunas,[1] Tom Morris,[2] Laura Gray[1]

## ABSTRACT

**Objectives** To investigate the extent to which cluster sizes vary in stepped-wedge cluster randomised trials (SW-CRT) and whether any variability is accounted for during the sample size calculation and analysis of these trials.

**Setting** Any, not limited to healthcare settings.

**Participants** Any taking part in an SW-CRT published up to March 2016.

**Primary and secondary outcome measures** The primary outcome is the variability in cluster sizes, measured by the coefficient of variation (CV) in cluster size. Secondary outcomes include the difference between the cluster sizes assumed during the sample size calculation and those observed during the trial, any reported variability in cluster sizes and whether the methods of sample size calculation and methods of analysis accounted for any variability in cluster sizes.

**Results** Of the 101 included SW-CRTs, 48% mentioned that the included clusters were known to vary in size, yet only 13% of these accounted for this during the calculation of the sample size. However, 69% of the trials did use a method of analysis appropriate for when clusters vary in size. Full trial reports were available for 53 trials. The CV was calculated for 23 of these: the median CV was 0.41 (IQR: 0.22–0.52). Actual cluster sizes could be compared with those assumed during the sample size calculation for 14 (26%) of the trial reports; the cluster sizes were between 29% and 480% of that which had been assumed.

**Conclusions** Cluster sizes often vary in SW-CRTs. Reporting of SW-CRTs also remains suboptimal. The effect of unequal cluster sizes on the statistical power of SW-CRTs needs further exploration and methods appropriate to studies with unequal cluster sizes need to be employed.

## BACKGROUND

Cluster randomised trials (CRT) are often used to evaluate healthcare interventions that are implemented at the cluster level, for example, within hospitals, general practices or across geographical areas, where treatment contamination would be likely to occur if individual randomisation was used.[1] In conventional parallel CRTs, clusters are generally randomised to either a control or intervention group (usually with equal numbers in each group). After baseline measurements have been taken,

[1]Department of Health Sciences, University of Leicester, Leicester, UK
[2]Leicester Clinical Trials Unit, University of Leicester, Leicester, UK

**Correspondence to**
Caroline Kristunas;
cak21@le.ac.uk

**Strengths and limitations of this study**

► To our knowledge, this is the first systematic review to assess how often and to what extent cluster sizes are unequal in stepped-wedge cluster randomised trials and determine whether the current methodology being used accounts for any variability in cluster sizes.
► This review used prespecified search and study selection strategies as well as double data extraction.
► Due to poor reporting quality, the actual cluster sizes could not be extracted for all studies, and cluster sizes were therefore estimated from whatever information was provided.

those randomised to the intervention group are exposed to the intervention for the remainder of the trial, whereas those in the control group are not.

The stepped-wedge CRT (SW-CRT) is a relatively recent variant of the CRT. In SW-CRTs, the clusters are randomised to start the intervention at different time points, known as steps, rather than being randomised to either a control or an intervention group. All of the clusters generally start the trial in a control group, then switch to the intervention when their time comes, until by the end of the trial all of the clusters are exposed to the intervention.[2] The cluster size is the number of individuals providing outcome measurements, either per cluster per measurement point, or per cluster over the whole duration of the trial. Figure 1 gives a schematic representation of an SW-CRT. Although use of this trial design is still relatively rare, recent systematic reviews have shown there to be a dramatic rise in the number of SW-CRTs published within the last few years.[3] However, the methodological advances for SW-CRTs lag behind those for other trial designs.

Due to the variability that occurs in the natural sizes of some clusters, the actual observed cluster sizes often vary within a

**Figure 1** Schematic representation of a stepped-wedge cluster randomised trial with four steps. The white areas represent control periods and the shaded areas represent intervention periods.

CRT.[1] For both cohort and cross-sectional designs, the actual observed cluster size can be defined as either the number of observations actually accrued over the whole duration of the trial or the number of observations actually accrued at each measurement point, with the latter enabling the former to be calculated. Clusters may also vary in size due to differences in the rates of loss to follow-up and incidence rates across clusters.[1] Increasing variability in cluster sizes has been shown to cause a loss of statistical power in parallel CRTs,[4] particularly if the coefficient of variation (CV) in cluster size (given as the SD of the cluster size divided by the mean cluster size) is greater than 0.23.[5] This can lead to a trial that is underpowered to detect the effect of the intervention that is being evaluated. Various sample size calculation methods have thus been developed for parallel CRTs which account for any imbalance in cluster sizes, to ensure that the trial will be sufficiently powered.[5–8] However, although the effect of unequal cluster sizes has been reported for parallel CRTs, it has only been reported for cross-sectional SW-CRTs with a continuous outcome and analysed using generalised estimating equations (GEE).[9] It is expected that some loss of power will be observed for certain subtypes of SW-CRTs and therefore a similar adjustment to the sample size calculation is likely to be required.

It has been recommended that mixed-effects models (MM) or GEEs should be used to analyse SW-CRTs.[10] In particular, linear MMs can be used for continuous data, plausibly normally distributed, or when cluster sizes are approximately equal, but if cluster sizes vary and the outcome is non-normal, then generalised linear MM (GLMM) or GEEs are preferred.[10] This is because both of these methods can account for the variability in cluster sizes, as well as non-normal data.[10] It is therefore to be expected that MMs and GEEs will be employed in SW-CRTs where the cluster sizes are likely to vary. Other methods of analysis that can also account for the design features of SW-CRTs (such as clustering and potential time effects) may also be suitable for use when cluster sizes vary; however, the appropriateness of these methods for the analysis of SW-CRTs with variable cluster sizes is currently unknown. To date, MMs and GEEs are the only methods of analysis for which the efficiency of the method has been investigated when cluster sizes vary.

Varying cluster sizes should also be considered during the randomisation process, as larger clusters may have different characteristics to smaller clusters. If more than one cluster switches to the intervention at each time point, then the randomisation process should be such that it ensures that cluster sizes are balanced across time points.

A recent systematic review of SW-CRTs found that few studies reported any variation in cluster sizes. However, major deficiencies were found in the reporting[11] and so this is likely to be an underestimation of the true extent of variability in cluster sizes. The size of the variability in cluster sizes in SW-CRTs has not previously been reported, nor how any imbalance in cluster sizes is being adjusted for during the calculation of the sample size or the analysis of these trials. The primary aims of our review were therefore to:

► determine how often and to what extent cluster sizes are unequal in SW-CRTs;
► determine how unequal cluster sizes are being taken into account during the calculation of the sample size and the analysis of SW-CRTs.

## METHODS
### Search strategy
We included in our review studies identified by a recent systematic review conducted by Martin *et al* of SW-CRTs up to 23 October 2014.[11] In addition, we

also included studies published more recently, by searching MEDLINE, Scopus and EBSCO Host, up to 21 March 2016, using an adapted version of the Martin *et al*'s search strategy (adapted to each database and to exclude studies prior to 23 October 2014). The search strategy is provided in online supplementary file 1. The studies that were included were from protocols or independent full reports of SW-CRTs in both healthcare and non-healthcare settings.[11] The studies had to be randomised trials, using cluster randomisation and have at least two steps. We did not restrict our search to trials where all clusters started in the control condition or ended in the intervention condition. Trials that were not published in English, were individually or non-randomised trials, trials with a crossover design and those that were retrospectively analysed as an SW-CRT were excluded.[11] We focused on original research studies and primary study reports.

Once duplicates had been removed, the titles and abstracts of the studies identified by the search were screened for eligibility by CAK. Full-text articles were then obtained for all potentially eligible studies and screened independently by two authors (CAK and TM). Those studies found not to meet the eligibility criteria were excluded and the reasons for exclusion tabulated. Any differences of opinion were resolved by discussions with LG.

### Data extraction

Data for all studies meeting the eligibility criteria were extracted by CAK and checked by TM. Any differences of opinion were resolved through discussions with all authors. A data extraction form was developed, then tested and refined on a small number of studies.

Basic trial characteristics are reported. These include the year of publication, type of cluster, the type of primary outcome (binary, continuous, time to event, and so on), number of clusters required and whether any restrictions related to cluster sizes were used in the randomisation procedure.

Information was also extracted in relation to variability in cluster sizes. We report whether it was mentioned in the report or protocol that clusters were known to vary in size prior to the trial commencing, whether any variability in cluster sizes was accounted for during the randomisation process or calculation of the sample size and whether the method of analysis appropriately accounts for this variability. We also recorded the number and sizes of the clusters assumed during the sample size calculation. For completed trials, we extracted the actual cluster sizes and reported the variability in cluster sizes that was observed during the trial, and made a comparison of the actual cluster sizes with the cluster sizes that were assumed during the sample size calculations.

Reporting of SW-CRTs is poor[11 12] and so to increase the number of trials available for a comparison of cluster sizes, some cluster sizes had to be estimated

from whatever information was available. Cluster sizes per time period, at a specific time point (such as at baseline) or the total cluster sizes over the full duration of the trial were preferred. If this information was not available then other information relating to the sizes of the clusters was used to estimate the cluster sizes, so that an estimate of the variability in the sizes of the clusters could be made. This included using the reported number of eligible individuals in each cluster as an approximation of the number of individuals included in each cluster, assuming that all clusters in a randomisation group (clusters randomised to switch to the intervention at the same time) were of equal size when only the mean cluster size per randomisation group was given, or using the mean and SD of cluster size in each randomisation group to find the CV for each group. The comparison of cluster sizes is reported separately for the actual cluster sizes and those that were estimated from summary measures.

### Analysis

We summarise the basic trial characteristics, methods of sample size calculation and methods of analysis, and number and size of clusters assumed during the sample size calculation for study reports and protocols. Actual cluster sizes, estimated actual cluster sizes and comparisons of actual cluster sizes with those that were assumed during the calculation of the sample size are summarised for trial reports only.

## RESULTS

We identified 371 records through our search of electronic databases. After combining this with the records identified by Martin *et al*,[11] removing three protocols for which the full reports had been identified, and after removing duplicates, we were left with 330 records. We screened these records and excluded 229 which did not meet our eligibility criteria (figure 2). A full list of the 101 included studies is given in online supplementary file 2.

The general characteristics of the included trials are presented in table 1. Full trial reports were available for 53 (52%) trials, with only protocols available for the remaining 48 (48%) trials. Two (2%) trials were conducted as pilot studies. A primary outcome was reported for 87 (86%) trials, of which the majority were binary (45%), time-to-event and rate (17%) or continuous (16%) outcomes, and six (7%) reported multiple primary outcomes. Three (3%) trials did not report a primary outcome and for the remaining 11 (11%) trials the primary outcome was not clear. For the majority of the trials (52%), the clusters comprised healthcare facilities, such as hospitals, hospital wards, clinics or general practitioner practices. The remaining clusters comprised geographical areas/communities (10%), residential/long-term care facilities (9%), healthcare professionals (9%), educational facilities

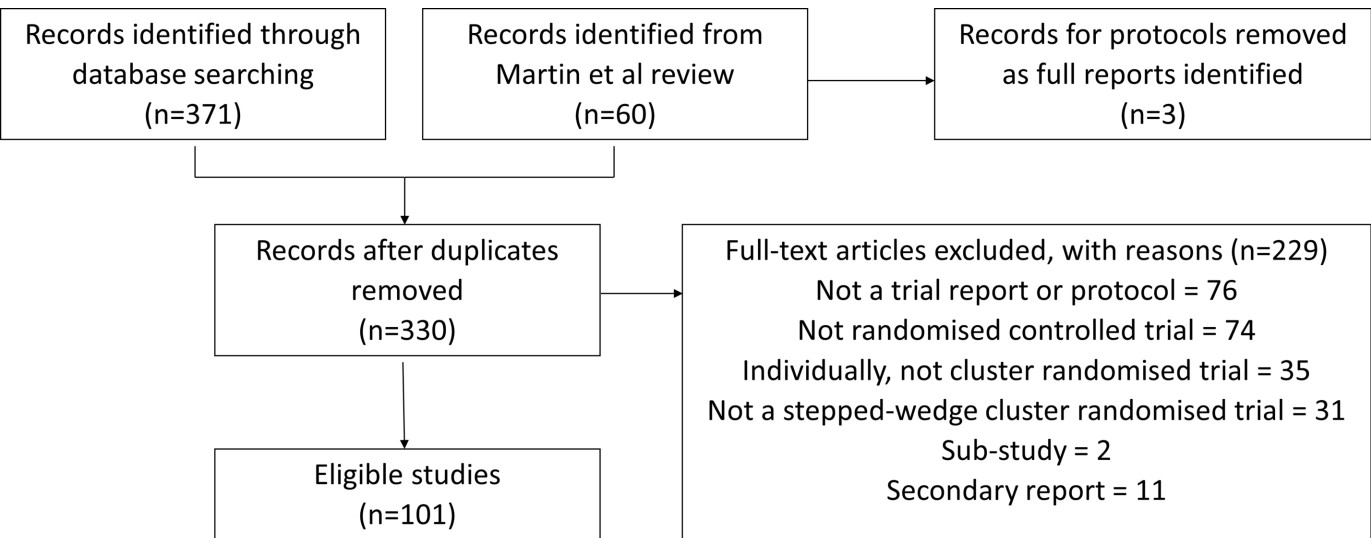

**Figure 2** Flow chart showing the studies identified by the systematic review.

(6%), rehabilitation clinics/programmes (5%), family groups/households (4%), or other groups (5%).

For just under half (48, 48%) of the trials there was evidence that the clusters were known to vary in size. Yet few of these accounted for inequality in cluster size during the randomisation process (29%) or sample size calculation (13%). Nine stratified by cluster size, one pair matched according to cluster size, two restricted randomisation by cluster size, one used minimisation to balance cluster sizes and one used block randomisation by cluster size. One additional trial used stratification, but did not state for which variables stratification was applied.

### Sample size calculation

Only 82 (81%) of the included studies presented a sample size calculation, of which 71 reported the number of clusters required and 57 reported the required cluster sizes (23 reported cluster size per time period, 34 reported total cluster size). The median (IQR) number of clusters required for these trials was 11 (6–22.5). The median (IQR) cluster size was 30 (20–125) for those that gave the cluster size per time period and 51 (20.75–165) for those that gave the total cluster size over the duration of the trial.

In total, only six (6%) trials reported that they had accounted for unequal cluster sizes when calculating the required sample size. Two used an estimate of the CV in cluster size, two used simulations to calculate the required sample size, varying the cluster sizes in their simulations, and for the remaining two trials the method used to account for unequal cluster sizes was unclear.

### Methods of analysis

A small proportion of the trials (12%) attempted to adjust for any inequality in cluster size through the inclusion of covariates in the analysis and one trial planned to conduct an individual-level analysis, rather

than cluster-level analysis, if clusters were found to vary in size.

The majority of analyses were conducted using MMs (56%), with 32 studies using GLMMs, 7 using linear MMs and 18 not distinguishing between the two. An equal number of trials used GEEs (13%) as used simple regression models (linear or generalised linear models) (13%). Only five of those that used simple regression models accounted for the clustering of the data during the analysis. The method of analysis for the primary outcome was unclear for five studies, and two studies did not report in their protocol details of how the analysis would be conducted. The remaining 11 (11%) studies employed simple analysis methods, such as analysis of variance and two sample t-tests, which did not account for the clustering of the data nor any period effect. The majority of studies (69%) employed a method of analysis known to be appropriate for when cluster sizes vary in SW-CRTs (LMM, GLMM or GEEs). Of the 48 studies for which there was evidence that cluster sizes were known to vary, 75.0% (36) of those used LMM, GLMM or GEEs in the analysis. However, for those 23 trials where the cluster sizes were actually found to vary in size, only 12 (52.2%) of them used LMM, GLMM or GEEs.

### Reporting of actual cluster sizes

Of the 53 included full trial reports, it was only possible to determine the actual cluster sizes for 13 (25%) of the trials. Five of these trials gave the cluster sizes during each time period, four gave the size of each cluster at a specific time point, such as at baseline or the point of analysis, and the remaining four reports gave the total size of each cluster over the duration of the whole trial. One further trial gave the number of eligible individuals in each cluster. A further eight (15%) reports gave a summary measure of the cluster sizes, such as the mean per month, per randomisation group, per intervention group, and so on, or the range of cluster sizes.

**Table 1** Trial characteristics of included SW-CRTs. Unless stated the denominator is the number of included studies (n=101)

| Characteristic | Number | Percentage |
|---|---|---|
| Included studies | 101 | |
| Publication type: | | |
| Report | 53 | 52.5 |
| Protocol | 48 | 47.5 |
| Publication describes results from a pilot study | 2 | 2.0 |
| Primary outcome reported | 87 | 86.1 |
| Primary outcome type (where reported, denominator=87): | | |
| Binary | 39 | 44.8 |
| Time-to-event and rate | 15 | 17.2 |
| Continuous | 14 | 16.1 |
| Count | 7 | 8.0 |
| Ordinal | 6 | 6.9 |
| Multiple | 6 | 6.9 |
| Types of cluster: | | |
| Health care facilities | 53 | 52.5 |
| Geographical areas/communities | 10 | 9.9 |
| Residential/long-term care facilities | 9 | 8.9 |
| Health care professionals (including groups thereof) | 9 | 8.9 |
| Education facilities | 6 | 5.9 |
| Rehabilitation clinics/programs | 5 | 5.0 |
| Family groups/households | 4 | 4.0 |
| Other | 5 | 5.0 |
| Evidence of clusters being known to vary in size | 48 | 47.5 |
| Sample size calculation presented | 82 | 81.2 |
| Accounted for unequal cluster sizes in sample size calculation | 6 | 5.9 |
| Method used to account for unequal cluster sizes in sample size calculation (where reported, denominator=6): | | |
| Used coefficient of variation in cluster size | 2 | 33.3 |
| Accounted for in simulation | 2 | 33.3 |
| Unclear | 2 | 33.3 |
| Reported number of clusters required | 71 | 70.3 |
| Median (interquartile range) number of clusters required (n=71) | 11 (6–22.5) | |
| Reported required cluster size | 57 | 56.4 |
| Median (interquartile range) per time period (n=23) | 30 (20-125) | |
| Median (interquartile range) overall (n=34) | 51 (20.75–165) | |
| Reported total sample size required* | 64 | 63.4 |
| Accounted for unequal cluster sizes in the randomisation | 29 | 28.7 |
| Method of analysis reported | 94 | 93.1 |
| Method of analysis used (where reported, denominator=94): | | |
| GEEs | 13 | 12.9 |
| MM | 57 | 56.4 |
| GLMM | 32 | 31.7 |
| LMM | 7 | 6.9 |
| Unclear | 18 | 17.8 |
| Other models | 16 | 15.8 |
| Generalised LM | 5 | 5.0 |

Continued

| Characteristic | Number | Percentage |
|---|---|---|
| GLM (accounting for clustering) | 5 | 5.0 |
| LM/ANCOVA | 6 | 5.9 |
| Simple analysis† | 8 | 7.9 |
| Unclear | 5 | 5.0 |
| Not given | 2 | 2.0 |

*Total sample size required either stated or enough detail given for it to be reproduced.
†Including McNemar's test, Mann-Whitney U test, t-test, analysis of variance, bivariate analysis and Wilcoxon signed-rank test.
ANCOVA, analysis of covariance; GEE, generalised estimating equation; GLM, generalised linear model; GLMM, generalised linear mixed-effects model; LM, linear model; LMM, linear mixed-effects model; MM, mixed-effects model; SW-CRT, stepped-wedge cluster randomised trial.

Two (4%) reports gave the sizes of the randomisation groups only.

### Comparison of cluster sizes

Few study reports presented both the required cluster sizes and the actual cluster sizes. The way in which cluster sizes were presented varied between studies and some only presented summary measures of the cluster sizes. Of the 24 trial reports that presented some indication of the cluster sizes, only 14 (58%) also stated the required cluster size. For those trials that presented the actual cluster sizes, the individual cluster sizes, either over the whole trial or at each time point, were between 29% and 480% of the required cluster size (median=100%, IQR=90%, 100%). For those trials that reported a summary measure of the actual cluster sizes, the cluster sizes were between 30% and 209% of the required cluster size (median=79%, IQR=65%, 101%). The average cluster size for each trial ranged from 66% to 214% of the required cluster size (median=95%, IQR=79%, 116%).

None of the trial reports presented the actual value of the CV in cluster size. It was possible to estimate the CV in cluster size for 23 of the trials, of which only two experienced no variability in cluster size. The median CV in cluster size was 0.41 (IQR: 0.22–0.52) and the maximum value was 1.29. None of these trials reported that they accounted for unequal cluster sizes during the sample size calculation.

### DISCUSSION

We conducted a systematic review to assess how often and to what extent cluster sizes are unequal in SW-CRTs, and to determine how any inequality in cluster sizes is currently being taken into account during the calculation of the sample sizes and the analysis of these trials. Of the 101 SW-CRTs included, almost half (48%) mentioned that the clusters were known to vary in size, yet many of these did not account for this during the calculation of the sample size. Simple analytical methods of sample size calculation which account for any variability in cluster sizes exist for parallel CRTs.[5–8] No such methods have been developed for use with SW-CRTs, though it may be possible to adapt the methods for parallel CRTs for use in SW-CRTs.

About 69% of the trials did however employ a method of analysis deemed appropriate for when cluster sizes are unequal in SW-CRTs (LMM, GLMM or GEEs).[10] There may be other methods of analysis that are also suitable for SW-CRTs with unequal cluster sizes, but these have yet to be investigated. In addition to the suitability of the analysis method to the variability in cluster sizes in an SW-CRT, the appropriateness of the method to the SW-CRT design must also be considered. Clustering and time effects will also need to be considered . Barker et al[13] and Davey et al[14] have both conducted reviews that consider the suitability of analysis methods for SW-CRTs.

Full trial reports were available for 53 trials. Only a quarter of these reported the sizes of each cluster, either per time period, at a specific point during the trial, or the total size over the whole duration of the trial. The CV in cluster size was calculated based on this information. A further eight trials reported a summary measure of the cluster sizes, such as the mean cluster size per month or the mean cluster size per randomisation group. From these summary measures, we were able to produce an estimate of the CV in cluster size for each of these trials. One further trial only gave the number of eligible individuals in each cluster and so, assuming equal recruitment rates in each cluster, it was also possible to estimate the CV in cluster size for this trial. Better reporting of CV is required for SW-CRTs, whether this be the CV across all periods or the CV per period.

Only two trials experienced no variability in cluster size. The majority (70%) of trials had a CV in cluster size >0.23, and so if these had been parallel CRTs we would have expected these trials to have suffered a significant loss of statistical power.[5] However, a recent simulation study has shown that for cross-sectional SW-CRTs with a continuous outcome and intracluster correlation coefficient (ICC) of 0.05, even high variability in cluster sizes does not cause a significant loss of power.[9] Nevertheless, a recent systematic review found that the majority of SW-CRTs are not of a cross-sectional

design,[12] and within health research an ICC of 0.05 is generally considered high. It is for these cohort, low ICC and other trials that the effect of unequal cluster sizes is still unknown. However, even if unequal cluster sizes are found to cause a reduction in power for most SW-CRTs, this effect may be overshadowed by the effect of using an incorrect sample size calculation, such as one that does not allow for the effect of time. Not allowing for the effect of time in this calculation can result in both underpowered and overpowered studies and so even with unequal cluster sizes, a study might still be overpowered.[11] It is therefore important that the sample size calculation for an SW-CRT correctly allows for all aspects of the design.

The actual cluster sizes also varied considerably from those that were assumed during the calculation of the required sample size, with reported cluster sizes being between 29% and 480% of that which was required. This would suggest that some of the trials we examined are likely to have been considerably overpowered or underpowered. However, half of the studies that presented the actual cluster sizes did use clusters within 10% of the required size and half of those that presented a summary measure of the actual cluster size used clusters within 22% of the actual size. These comparisons could only be made for 14 (26%) of the trials with published reports. Information both on the sizes of the clusters that were assumed during the calculation of the sample size and the actual cluster sizes that were observed during the trial was often not reported.

The majority of systematic reviews of SW-CRTs have been conducted with the aim of assessing the quality and breadth of SW-CRTs.[3 15 16] A couple of recent systematic reviews have also been conducted to assess the quality of reporting and methodological rigour of SW-CRTs,[11 12] as well as the statistical methodology being used and available for these trials.[13] However, none of these systematic reviews have assessed the prevalence and severity of unequal cluster sizes in SW-CRTs, nor the statistical methodology being used to account for this variability in cluster sizes. Our review has highlighted the high degree of variability in cluster sizes that is observed in SW-CRTs. Knowing that cluster sizes often vary in SW-CRTs can allow researchers to plan for an expected inequality in cluster sizes, taking this into account when calculating the required sample size for these trials and choosing an appropriate method of analysis. We also highlight the need for improved quality of reporting of cluster sizes in SW-CRTs, which concurs with the findings of previous reviews[11 12] and the need for wider implementation of methodologies which account for unequal cluster sizes. These findings are in line with what has been shown more generally in CRTs.[17]

Our study has several strengths. In order to minimise the potential for bias during the review, we prespecified search and study selection strategies. We also did not limit our search to publications from journals with a high impact factor, increasing the generalisability of

our findings. However, our study does have limitations as well. For example, selection bias may have been introduced by the choice to only include studies published in English. It was also difficult to ensure the identification of all SW-CRTs, as many do not include in either their title or abstract the common terms that identify an SW-CRT. We included in our search strategy all of the common identifiers that have been included in previous reviews, yet this is still not sufficient to identify studies where the researchers were not aware that their studies were SW-CRTs and therefore did not include common identifiers in their title or abstract. This may introduce some selection bias. The poor reporting of SW-CRTs[11 12] may limit the generalisability of the findings of our study. It was only possible to make a comparison of cluster sizes and to calculate the CV in cluster size for a small number of studies, and these are likely to be those studies that have a better quality of reporting than the other studies. The estimations that were made when determining the observed cluster sizes may also result in an underestimation or overestimation of the degree of variability in cluster sizes and the comparability of the observed cluster sizes with those which were required.

Unequal cluster sizes are a common problem in SW-CRTs. Variability in cluster sizes is not being accounted for during the calculation of the sample size for these trials and the reporting of the actual cluster sizes observed during these trials is poor, which limits the scope of this review. Appropriate methods of sample size calculation that account for variability in cluster sizes should be developed by methodological statisticians and perhaps implemented in appropriate software in order to allow their widespread use. Simulation methods, such as those used by Baio et al,[18] could be used while simple analytical methods are developed. Further research is also required to investigate the effect of unequal cluster sizes on the statistical power for cohort SW-CRTs and those with a binary outcome. An investigation into how our results differ between cohort and cross-sectional designs would also be of interest.

We echo the views of others,[2] that the development of reporting guidelines for SW-CRTs is needed, particularly to improve the suboptimal quality of reporting for the sample size calculations and of the actual observed cluster sizes in SW-CRTs. In the meantime, it is recommended that reporting should follow the CONSORT (Consolidated Standards of Reporting Trials) 2010 extension to CRTs[19] and the modifications presented by Hemming et al in their paper.[2] A CONSORT extension to SW-CRTs is currently in development.[20]

## CONCLUSIONS

Cluster sizes were found to vary in SW-CRTs. The effect of unequal cluster sizes on SW-CRTs is yet to be reported for cohort designs and those with a binary outcome, and so the effect of this observed variability

is currently unknown. Methods of sample size calculation that are used for these trials rarely account for any variability in cluster size. Reporting of cluster sizes in both the reported sample size calculation and those actually observed during the trial is poor. Authors need to provide greater detail of how the required sample size has been calculated as well as providing details of the actual cluster sizes observed during the trial, preferably at each step of the trial. Researchers should also be aware that cluster sizes are likely to vary in SW-CRTs and so sample size calculation methods and analysis methods should be used which can account for this variability.

**Contributors** LG conceptualised the review. CAK carried out the systematic review, designed the data extraction form, identified the studies for inclusion, extracted data, performed the data analysis and wrote the first draft. TM carried out the second data extraction and critically reviewed the contents of the paper. LG resolved any differences between CAK and TM in the data extraction. All authors contributed to the writing of the paper, and read and approved the final manuscript.

**Funding** CK is funded by a National Institute for Health Research (NIHR) Research Methods Fellowship (NIHR-RMFI-2014-05-019). The authors would also like to acknowledge support from the NIHR Collaboration for Leadership in Applied Health Research and Care – East Midlands (NIHR 384 CLAHRC – EM), Leicester Clinical Trials Unit and the NIHR Leicester-Loughborough Diet, Lifestyle and Physical Activity Biomedical Research Unit, which is a partnership between University Hospitals of Leicester NHS Trust, Loughborough University and the University of Leicester. The views expressed are those of the authors and not necessarily those of the NIHR or the Department of Health.

**Competing interests** None declared.

**Provenance and peer review** Not commissioned; externally peer reviewed.

**Data sharing statement** There are no further data available.

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
