## [Reviewer comments · BMJ Open]

ARTICLE DETAILS

TITLE (PROVISIONAL)	Unequal cluster sizes in stepped-wedge cluster randomised trials: a systematic review
AUTHORS	Kristunas, Caroline; Morris, Tom; Gray, Laura

VERSION 1 – REVIEW

REVIEWER	Mona Kanaan University of York, UK
REVIEW RETURNED	05-May-2017

GENERAL COMMENTS	This manuscript is a systematic review of the reporting of unequal cluster sizes in stepped-wedge cluster randomised trials. In the main, the article is well written and executed. However, there are a number of issues that the authors might want to address. I first list the major points then the minor ones. - In the abstract and in various places in the manuscript extreme care should be taken in terms of drawing the parallel between SW_CRT and parallel CRTs in terms of loss-of-power when unequal cluster sizes are not accounted for in the analysis and the sample size calculations. I would suggest removing this comparison from the abstract and in the main text minimising the number of times this point is made; as the research to investigate this impact in the case of SW-CRT is yet to be conducted.- The methodology used for estimating the cluster size when it is not reported should be clearly specified.- In Table 1, it should be clear which denominator is being used. For example, in some instances the percentage refers to the percentage of the total sample size, whereas in others, it refers to the percentage within studies that reported that characteristic. Also, for each characteristic the total number of studies that report that characteristic should be clearly specified. In addition, Table 1 title should reflect what is presented in the table; summary statistics other than percentages were reported in the Table.- The methods of analyses are reported in Table 1, however, I would have liked to see as well whether these methods were fit for purpose and whether they accounted for the study design appropriately.- Care should be taken when speaking about under- or over-powered results based only on issues related to the cluster size; as the cluster size is only one element of sample size calculations for these studies.
--

	Minor Points:  - Specify how “gave some indication” was assessed. - P2 L44/45 could you please clarify the sentence “Some cluster sizes could only be estimated due to the quality of the reporting of some studies.” - P5 L157/8 the numbers here do not tie up with the numbers in Table 1. In Table 1, it seems that rate and continuous were lumped together whereas in the text they are individually reported. - P6 L168 give the absolute number in addition to the percentage - P7 L177/178 for the text enclosed within brackets add “reported” after the two numbers. - P7 L178/181 suggest to split into two sentences. - P10 L305 replace “or cohort” with “for cohort” - P16 onwards provide a title for this list of references
--	--

REVIEWER	Michael J. Grayling MRC Biostatistics Unit, UK
REVIEW RETURNED	11-Jun-2017

GENERAL COMMENTS	This manuscript explores the variability of cluster sizes in stepped-wedge (SW) cluster randomised trials (CRTs). Principally, the cluster sizes assumed at the design stage of trials are compared to those ultimately realised, and implications on trial design and analysis are discussed. This is an important issue in SW-CRTs that to date has not received as much attention as it should have. Consequently, this article contains results of substantial contemporary interest; highlighting the need for researchers to more adequately address cluster size variability in both SW-CRT design and analysis. I believe this work warrants publication in BMJ Open; I have few major concerns over its content, but I do feel the article could be improved by taking in to account the comments given below. 1. Major Comments 1.1. The major focus of the article is on “cluster sizes”, however no clear definition of what is meant by cluster size is included. For those not familiar with SW-CRTs this will be particularly confusing. For example, what is the cluster size in a cross-sectional SW-CRT relative to a cohort design? 1.2. You state that GLMMs or GEEs are the preferred methods of analysis when cluster sizes vary. However, this is only the case for non-normal endpoints.
--

Hussey and Hughes (2007) discussed how LMMs could be used for normal data with varying cluster size, or for non-normal data if the cluster sizes were similar. Moreover, GLMs that account for clustering, or arguably even other approaches, can all potentially be fine methods of analysis. Therefore, several of the figures you present on the analysis methods I feel need to be revised. There is also a question over whether you want to assess if the analysis was pre-specified appropriately (i.e., the authors either expected or did not expect cluster variability and chose an appropriate method for their analysis), specified correctly post data accrual, or simply present overall figures on what method of analysis was used. You should thus revise the manuscript to make it clear what it is you are trying to show in regards to the utilised analysis methods, and present figures for this.

2. Minor Comments

2.1. Related to comment 1.1, clarification needs to be provided as to what is meant by “observed cluster sizes” (L69). Is this the number of observations actually accrued?

2.2. Results should be reported consistently with regards to the IQR and range (L175-186 and L224).

2.3. A clearer description of what studies were included should be given (L99). For example, where studies with cluster starting in the intervention or ending in the control condition included?

2.4. It only really becomes clear in the results section what you meant by “we then updated this review” (L101). Could you please add a sentence in the Methods section to make it obvious you planned to include those studies from Martin et al (2016), and then add additional more recent papers via your search.

2.5. L110: Who performed the initial screening? Was this done by a single person?

2.6. Where the search strategies the same for each database (L102)? Or was a single search performed via one website across the databases?

2.7. All of the exclusion criteria are not obvious to me relative to the reasons given in the flow diagram. For example, what is a “qualitative report” and why was this therefore excluded?

2.8. What is meant by “per randomisation group” on L139?

	2.9. L83 and L192: MM has been used twice with two different meanings. 2.10. There are several instances where you state you will explore “calculation of sample size or analysis”. I think these should be revised to make it clear that this isn’t a composite exploration; you are assessing methods of sample size calculation and methods of analysis separately. 2.11. In the discussion you state that “no such methods” exist for accounting for cluster variability (L237). However, simulation is obviously available for this, there is just not simple analytical approaches available for all types of SW-CRT. Could you please clarify this sentence to this effect. 2.12. You conclude the discussion by saying “reporting guidelines” need to be developed (L306). It would be good to reference the CONSORT extension via its protocol here. 2.13. L312: “several designs” does not make sense to me. I suggest expanding this to make it clear what types of SW-CRT there is no method available for. 2.14. I would suggest revising a few of the references or sentences to make it clear that Martin et al (2016) refers to poor reporting of sample size calculations, and Grayling et al (2017) is general poor reporting. 3. Typographical/Grammatical The authors may find the following suggested corrections sensible: 3.1. L67: “designs of trial” should be “trial designs” I think. 3.2. L81: “SW-CRTs of some designs” should be “certain sub-types of SW-CRT” or similar. 3.3. L108: “focussed” should be “focused”. 3.4. L285: “don’t” should be “do not”. 3.5. L305: “Or” should be “of”.
--	--

REVIEWER	Fan Li Department of Biostatistics and Bioinformatics, Duke University, U.S.A.
REVIEW RETURNED	18-Jun-2017

GENERAL COMMENTS	Unequal cluster sizes are frequently seen in parallel cluster-randomized trials (CRTs), and have also been increasingly observed in stepped-wedge (SW) CRTs. The authors reviewed the frequency and the extent to which cluster sizes varied in practice and the appropriateness of related design and analysis strategies being used. This is an important topic given the rising interest in using stepped-wedge designs and several unresolved issues on the statistical implications of unequal cluster sizes. This is a generally well-written paper, and I have the following suggestions in preparing the final version for publication.  1. Could the authors comment on how the review results (frequency and the extent to which cluster sizes vary) differ (or not differ) by cohort and cross-sectional designs? Several existing review papers suggest that the cohort designs are at least as prevalent as cross-sectional designs in stepped wedge studies. It may not be possible to present the results separately for cohort and cross-sectional studies since we may not be able to identify the type of each study reviewed (because of insufficient information), but comments based on your findings would be helpful. 2. Unlike in parallel CRTs, the coefficient of variation (CV) of cluster sizes in stepped-wedge CRTs can either quantify the variation in cluster size (# of participants) across all periods or the variation in the cluster size per period, as the author noted on page 8. Given that the current practice of reporting CV is relatively inadequate, could the author comment on the choice of which CV to report in practice? 3. On page 3, line 84-86, the authors seem to suggest that linear mixed models are not preferred when the cluster sizes vary. Could the authors be clearer about this point? It seems that the linear mixed models used by Hussey and Hughes are based on the cluster-period means, while in practice we could use linear mixed model at the individual level, just like GEE and GLMM.
---

REVIEWER	Mike Campbell University of Sheffield UK
REVIEW RETURNED	19-Jun-2017

GENERAL COMMENTS	This is one of many reviews published recently which identifies issues with the current design, analysis and reporting of cluster stepped wedge designs.(SWDs) In my view it just find another rather specialised area where they fail, does not add much to the general view that SWD's are badly designed and improvements via the upcoming CONSORT statement are needed. 1)I think the authors could discuss the earlier reviews in more detail, particularly to what extent the papers they considered overlap with their own review, since there are few SWD trials published as yet and so some have been examined in detail already a number of times
--

	2) The results from reference 4 in the paper have been shown to exaggerate the loss of power (See ref 1 &2). Thus I suspect the problem is less severe than the authors make out. 3) The actual effect of varying cluster sizes could be explored using the simulation program provided by Baio et al (ref 3) (I think) Refs 1 Van Breukelen GJP and Candel MJJM (2012). Comments on "Efficiency loss because of varying cluster size in cluster randomized trials is smaller than literature suggests". Statistics in Medicine, 31: 397-400 2 Campbell MJ and Walters SJ (2014) How to design, analyse and report cluster randomised trials in medicine and health related research. Wiley-Blackwell 247pp ISBN 978-1-119-99202-8 (Page 70-71) 3) Baio G, Copas A, Ambler G, et al. Sample size calculation for a stepped wedge trial, Trials 362 2015;16:354
--	---

REVIEWER	Jennifer Thompson London School of Hygiene & Tropical Medicine Both myself and the main author are part of a larger group developing CONSORT guidelines for stepped-wedge trials.
REVIEW RETURNED	23-Jun-2017

GENERAL COMMENTS	This paper is a well-conducted systematic review looking at whether the cluster size of SW-CRTs vary and to what extent they vary. The paper looks at whether the trials used methods appropriate for unequal cluster sizes where necessary. This is a good paper, addressing a gap in the literature of SW-CRTs. My comments focus mainly on areas where the paper needs improvements in clarity or consistency in the reporting. Essential The primary outcome stated in the abstract does not match up with the aims stated at the end of the background section, please change to reflect the main paper. The result in the abstract that 45% of trials with vary cluster size used an "appropriate" analysis is not reported in the main paper, please add this. At the bottom of the third paragraph of the background, the paper by Baio et al is given as a reference for an expectation that there will be a loss of power from unequal cluster sizes in SW-CRTs. Baio did not look at the effect of unequal cluster sizes on power so this reference does not support this statement. Please either find a different reference or edit the statement to reflect that it is not supported by any literature. It is unclear in the paper what is meant by an appropriate analysis with unequal cluster sizes, please clarify this. Some analysis methods are less efficient with unequal cluster sizes, but as far as I know they still give valid estimates and inference, so may be chosen despite their inefficiency. For example, in the fourth paragraph of the background, you should clarify that the methods you state are preferred when cluster sizes are unequal because they are more efficient.
---

It is also unclear which of the analysis methods in the results are “appropriate” or “inappropriate” for unequal cluster sizes, and which were from trials with unequal cluster sizes. This makes any interpretation of the analysis methods results difficult. Please add some comments on which methods were used with equal vs unequal clusters, and which were “appropriate” for this.

Please use a different measure of the spread of results throughout; the range isn’t very informative. An interquartile range would be more informative.

In table one, the row name “rate” is on the same line as “continuous”. I think continuous is meant to be on the next line. What is the difference between a rate outcome and a time to event outcome? Similarly, what is the difference between a proportion outcome and a binary outcome? What do you mean by a discrete outcome? Please clarify.

On page 8 paragraph 1, please give a median or mean for the difference in the actual and required sample size, as well as a measure of the spread.

It would be informative to add details of how you estimated cluster size where the paper gave partial information.

Optional

Since you report the proportion that accounted for unequal cluster size in the randomisation, you could add something in the background about why and when this should be considered. You could add whether studies accounted for varying cluster size in the randomisation to table one.

Page 7, methods of analysis section: you could add that the simple analysis methods also fail to adjust for period effects as well as failing to account for clustering.

In the discussion, you highlight that the paper for continuous cross-sectional designs is limited to cross-sectional designs. You could also highlight that this only considered an ICC of 0.05; this is high for health research. Unequal cluster sizes may have a larger impact when the ICC is lower.

VERSION 1 – AUTHOR RESPONSE

Reviewer: 1

Reviewer Name: Mona Kanaan

Institution and Country: University of York, UK. Please state any competing interests: None Declared

Please leave your comments for the authors below

This manuscript is a systematic review of the reporting of unequal cluster sizes in stepped-wedge cluster randomised trials. In the main, the article is well written and executed.

Comment: However, there are a number of issues that the authors might want to address. I first list the major points then the minor ones.

Response: *Thank you for your kind feedback and useful suggestions. Each of your comments is individually addressed below.

Comment: In the abstract and in various places in the manuscript extreme care should be taken in terms of drawing the parallel between SW_CRT and parallel CRTs in terms of loss-of-power when unequal cluster sizes are not accounted for in the analysis and the sample size calculations. I would suggest removing this comparison from the abstract and in the main text minimising the number of times this point is made; as the research to investigate this impact in the case of SW-CRT is yet to be conducted.

Response: We have removed this comparison from the abstract and from several places within the main text and in addition we have made it clearer in the Discussion that the effect of unequal cluster sizes on SW-CRTs is unknown.

Comment: The methodology used for estimating the cluster size when it is not reported should be clearly specified.

Response: Thank you for bringing this to our attention. A more detailed description of the methodology used for estimating the cluster sizes when it is not reported has been added to the Methods.

Comment: In Table 1, it should be clear which denominator is being used. For example, in some instances the percentage refers to the percentage of the total sample size, whereas in others, it refers to the percentage within studies that reported that characteristic. Also, for each characteristic the total number of studies that report that characteristic should be clearly specified. In addition, Table 1 title should reflect what is presented in the table; summary statistics other than percentages were reported in the Table.

Response: Changes have been made to the title and content of Table 1 to make it clearer how the values have been calculated and so that the title reflects the content of the table.

Comment: The methods of analyses are reported in Table 1, however, I would have liked to see as well whether these methods were fit for purpose and whether they accounted for the study design appropriately.

Response: Barker et al [1] and Davey et al [2] have both published reviews in which the appropriateness of the methods of analysis that have been used in published SW-CRTs have been discussed. We have therefore restricted our review to looking at whether the method used has been shown to be appropriate for when cluster sizes vary. We have clarified which methods we consider appropriate for varying cluster sizes (LMM, GLMM and GEEs). Discussion of whether those trials with varying cluster sizes used appropriate methods of analysis can be found in the results.

[1] Barker, D., et al. "Stepped wedge cluster randomised trials: a review of the statistical methodology used and available." *BMC medical research methodology* 16.1 (2016): 69.

[2] Davey, Calum, et al. "Analysis and reporting of stepped wedge randomised controlled trials: synthesis and critical appraisal of published studies, 2010 to 2014." *Trials* 16.1 (2015): 358.

Comment: Care should be taken when speaking about under- or over-powered results based only on issues related to the cluster size; as the cluster size is only one element of sample size calculations for these studies.

Response: A few sentences have been added to the discussion that stress that cluster size is only one element of sample size calculations and that even allowing for unequal cluster sizes, if the sample size calculation does not correctly allow for other aspects of the design, then the study will still be incorrectly powered.

Minor Points:

Comment: Specify how "gave some indication" was assessed.

Response: Changes have been made in the text to make it clearer how this "indication" was assessed.

Comment: P2 L44/45 could you please clarify the sentence "Some cluster sizes could only be estimated due to the quality of the reporting of some studies."

Response: This sentence has been rephrased to clarify its meaning.

Comment: P5 L157/8 the numbers here do not tie up with the numbers in Table 1. In Table 1, it seems that rate and continuous were lumped together whereas in the text they are individually reported.

Response: Thank you. An error occurred in the table and "continuous" should have appeared on the line below. This has now been corrected. Bearing this in mind, we believe that the numbers in the text do now tie up with the numbers in Table 1.

Comment: P6 L168 give the absolute number in addition to the percentage

Response: The absolute numbers have been added.

Comment: P7 L177/178 for the text enclosed within brackets add "reported" after the two numbers.

Response: Thank you, "reported" has now been added after each number.

Comment: P7 L178/181 suggest to split into two sentences.

Response: This sentence has been split into two sentences.

Comment: P10 L305 replace “or cohort” with “for cohort”

Response: This typo has been corrected.

Comment: P16 onwards provide a title for this list of references

Response: A title has been added to the list of references.

Reviewer: 2

Reviewer Name: Michael J. Grayling

Institution and Country: MRC Biostatistics Unit, UK Please state any competing interests: None declared

Please leave your comments for the authors below Please see attached.

Summary

This manuscript explores the variability of cluster sizes in stepped-wedge (SW) cluster randomised trials (CRTs). Principally, the cluster sizes assumed at the design stage of trials are compared to those ultimately realised, and implications on trial design and analysis are discussed. This is an important issue in SW-CRTs that to date has not received as much attention as it should have.

Consequently, this article contains results of substantial contemporary interest; highlighting the need for researchers to more adequately address cluster size variability in both SW-CRT design and analysis. I believe this work warrants publication in BMJ Open; I have few major concerns over its content, but I do feel the article could be improved by taking in to account the comments given below.

Response: Thank you for your kind feedback and useful suggestions. Each of your comments are addressed in detail below.

1. Major Comments

1.1. The major focus of the article is on “cluster sizes”, however no clear definition of what is meant by cluster size is included. For those not familiar with SW-CRTs this will be particularly confusing. For example, what is the cluster size in a cross-sectional SW-CRT relative to a cohort design?

Response: Cluster size for SW-CRTs has now been defined in the background.

1.2. You state that GLMMs or GEEs are the preferred methods of analysis when cluster sizes vary. However, this is only the case for non-normal endpoints. Hussey and Hughes (2007) discussed how LMMs could be used for normal data with varying cluster size, or for non-normal data if the cluster sizes were similar. Moreover, GLMs that account for clustering, or arguably even other approaches, can all potentially be fine methods of analysis. Therefore, several of the figures you present on the analysis methods I feel need to be revised. There is also a question over whether you want to assess if the analysis was pre-specified appropriately (i.e., the authors either expected or did not expect cluster variability and chose an appropriate method for their analysis), specified correctly post data accrual, or simply present overall figures on what method of analysis was used. You should thus revise the manuscript to make it clear what it is you are trying to show in regards to the utilised analysis methods, and present figures for this.

Response: Changes have been made to clarify that in some cases LMMs are also appropriate when cluster sizes vary. Although other methods of analysis might potentially be fine, only LMMs, GLMMs and GEEs have been investigated to date with varying cluster sizes, and therefore we can currently only endorse only these methods when clusters vary in size. Figures relating to the analysis methods have been changed to include LMMs as an appropriate method. A sentence has also been included that states that other methods may also be appropriate but that they have not yet been investigated. It has been clarified that we assess the analysis method in both the context of studies that indicated that clusters were known to vary in size, as well as those studies where clusters were shown to vary in size. Figures for these have been added.

2. Minor Comments

2.1. Related to comment 1.1, clarification needs to be provided as to what is meant by “observed cluster sizes” (L69). Is this the number of observations actually accrued?

Response: Thank you for bringing this to our attention. A description of what is meant by “observed cluster sizes” has been added.

2.2. Results should be reported consistently with regards to the IQR and range (L175-186 and L224).

Response: All ranges have been changed to interquartile ranges.

2.3. A clearer description of what studies were included should be given (L99). For example, were studies with clusters starting in the intervention or ending in the control condition included?

Response: We did not restrict our search to studies where all clusters started in the control condition and ended in the intervention condition. We have added a sentence to clarify this in the methods.

2.4. It only really becomes clear in the results section what you meant by “we then updated this review” (L101). Could you please add a sentence in the Methods section to make it obvious you planned to include those studies from Martin et al (2016), and then add additional more recent papers via your search.

Response: Thank you for making us aware that this was not immediately clear. The relevant section in the methods has been reworded for clarification.

2.5. L110: Who performed the initial screening? Was this done by a single person?

Response: The initial screening was conducted by the lead author (CK) and this detail has been added to the methods.

2.6. Were the search strategies the same for each database (L102)? Or was a single search performed via one website across the databases?

Response: Each database was searched separately with the same search strategy, but adapted slightly to the format of each database. This has been clarified in the methods.

2.7. All of the exclusion criteria are not obvious to me relative to the reasons given in the flow diagram. For example, what is a “qualitative report” and why was this therefore excluded?

Response: The excluded study was a report of the qualitative findings of a non-randomised trial and so should have been included under “non-randomised” as its reason for exclusion. The flow diagram has been changed accordingly.

2.8. What is meant by “per randomisation group” on L139?

Response: The phrase “per randomisation group” has been explained in the methods (lines 109 and 168 of the marked copy).

2.9. L83 and L192: MM has been used twice with two different meanings.

Response: Thank you for bringing this to our attention. Multilevel was written by mistake and has been replaced by mixed-effects.

2.10. There are several instances where you state you will explore “calculation of sample size or analysis”. I think these should be revised to make it clear that this isn’t a composite exploration; you are assessing methods of sample size calculation and methods of analysis separately.

Response: Changes have been made in a couple of places in the text of the methods for clarification.

2.11. In the discussion you state that “no such methods” exist for accounting for cluster variability (L237). However, simulation is obviously available for this, there is just not simple analytical approaches available for all types of SW-CRT. Could you please clarify this sentence to this effect.

Response: This sentence has been clarified and now states that there are no “simple analytical methods” that exist. Later in the discussion we now mention that simulations can also be used.

2.12. You conclude the discussion by saying “reporting guidelines” need to be developed (L306). It would be good to reference the CONSORT extension via its protocol here.

Response: The CONSORT extension protocol is now referenced here.

2.13. L312: “several designs” does not make sense to me. I suggest expanding this to make it clear what types of SW-CRT there is no method available for.

Response: “Several designs” has been replaced in this sentence with examples such as cohort designs and those with binary outcomes.

2.14. I would suggest revising a few of the references or sentences to make it clear that Martin et al (2016) refers to poor reporting of sample size calculations, and Grayling et al (2017) is general poor reporting.

Response: We have changed some of the references to clarify the differences in these reviews.

3. Typographical/Grammatical

Response: The authors may find the following suggested corrections sensible:

3.1. L67: “designs of trial” should be “trial designs” I think.

Response: Thank you, “designs of trial” has been replaced with “trial designs”.

3.2. L81: "SW-CRTs of some designs" should be "certain sub-types of SW-CRT" or similar.

Response: "SW-CRTs of some designs" has been replaced with "certain sub-types of SW-CRT".

3.3. L108: "focussed" should be "focused".

Response: "Focussed" has been replaced with "focused".

3.4. L285: "don't" should be "do not".

Response: Thank you, "don't" has been replaced with "do not".

3.5. L305: "Or" should be "of".

Response: This typo has been corrected.

Reviewer: 3

Reviewer Name: Fan Li

Institution and Country: Department of Biostatistics and Bioinformatics, Duke University, U.S.A.

Please state any competing interests: None declared

Please leave your comments for the authors below

Comment: Unequal cluster sizes are frequently seen in parallel cluster-randomized trials (CRTs), and have also been increasingly observed in stepped-wedge (SW) CRTs. The authors reviewed the frequency and the extent to which cluster sizes varied in practice and the appropriateness of related design and analysis strategies being used. This is an important topic given the rising interest in using stepped-wedge designs and several unresolved issues on the statistical implications of unequal cluster sizes. This is a generally well-written paper, and I have the following suggestions in preparing the final version for publication.

Response: Thank you for your kind and helpful comments, they have helped us to improve the paper. We have responded to each comment individually below.

Comment 1. Could the authors comment on how the review results (frequency and the extent to which cluster sizes vary) differ (or not differ) by cohort and cross-sectional designs? Several existing review papers suggest that the cohort designs are at least as prevalent as cross-sectional designs in stepped wedge studies. It may not be possible to present the results separately for cohort and cross-sectional studies since we may not be able to identify the type of each study reviewed (because of insufficient information), but comments based on your findings would be helpful.

Response: We would very much like to be able to comment on the differences between the cohort and cross-sectional designs, but unfortunately, insufficient information was given to allow us to comprehensively assess the extent to which cluster sizes vary by cohort and cross-sectional designs. We therefore feel that it would be inappropriate for us to make comments on this issue in the paper. We have added to the discussion the need for further research to investigate this.

Comment 2. Unlike in parallel CRTs, the coefficient of variation (CV) of cluster sizes in stepped-wedge CRTs can either quantify the variation in cluster size (# of participants) across all periods or the variation in the cluster size per period, as the author noted on page 8. Given that the current practice of reporting CV is relatively inadequate, could the author comment on the choice of which CV to report in practice?

Response: We have added a sentence encouraging the reporting of the coefficient of variation in cluster size. Although reporting of both CVs (across all periods and per period) would be beneficial in order to understand the variability in cluster sizes, we feel that reporting of either CV is still useful and would therefore not recommend the reporting of one CV over the other.

Comment 3. On page 3, line 84-86, the authors seem to suggest that linear mixed models are not preferred when the cluster sizes vary. Could the authors be clearer about this point? It seems that the linear mixed models used by Hussey and Hughes are based on the cluster-period means, while in practice we could use linear mixed model at the individual level, just like GEE and GLMM.

Response: Changes have been made to show that LMMs are also an appropriate method of analysis and the criteria used for determining when an appropriate method of analysis has been used have been clarified.

Reviewer: 4

Reviewer Name: Mike Campbell

Institution and Country: University of Sheffield, UK Please state any competing interests: None declared

Please leave your comments for the authors below

This is one of many reviews published recently which identifies issues with the current design, analysis and reporting of cluster stepped wedge designs.(SWDs) In my view it just find another rather specialised area where they fail, does not add much to the general view that SWD's are badly designed and improvements via the upcoming CONSORT statement are needed.

Comment 1) I think the authors could discuss the earlier reviews in more detail, particularly to what extent the papers they considered overlap with their own review, since there are few SWD trials published as yet and so some have been examined in detail already a number of times

Response: Thank you for your comment. Our review includes all of the studies included in the Martin et al. [3] review, but has been updated to include more recent studies as well. We have changed the wording in the methods to clarify this. Although many of the studies that have been included in our review will also have been included in many of the other reviews, the strengths and limitations of this study section and a paragraph in the discussion explain that our review is quite different to these in its objectives. Previous reviews have focussed on assessing the quality and breadth of SW-CRTs or the statistical methodology being used or available for these trials, whereas our review examines the prevalence and severity of unequal cluster sizes in SW-CRTs.

[3] Martin, James, et al. "Systematic review finds major deficiencies in sample size methodology and reporting for stepped-wedge cluster randomised trials." *BMJ open* 6.2 (2016): e010166.

Comment 2) The results from reference 4 in the paper have been shown to exaggerate the loss of power (See ref 1 &2). Thus I suspect the problem is less severe than the authors make out.

Response: Most references to this result has been removed from this paper.

Comment 3) The actual effect of varying cluster sizes could be explored using the simulation program provided by Baio et al (ref 3) (I think)

Response: Thank you for making us aware of this oversight. A sentence has been added to the methods to make the reader aware that simulation methods, such as those used by Baio et al., can be used to investigate the actual effect of varying cluster sizes.

Refs

1 Van Breukelen GJP and Candel MJJM (2012). Comments on “Efficiency loss because of varying cluster size in cluster randomized trials is smaller than literature suggests”. *Statistics in Medicine*, 31: 397-400

2 Campbell MJ and Walters SJ (2014) How to design, analyse and report cluster randomised trials in medicine and health related research. Wiley-Blackwell 247pp ISBN 978-1-119-99202-8 (Page 70-71)

3) Baio G, Copas A, Ambler G, et al. Sample size calculation for a stepped wedge trial, *Trials* 362 2015;16:354

Reviewer: 5

Reviewer Name: Jennifer Thompson

Institution and Country: London School of Hygiene & Tropical Medicine Please state any competing interests: Both myself and the main author are part of a larger group developing CONSORT guidelines for stepped-wedge trials.

Please leave your comments for the authors below

Please find my comments in the attached document.

Comment: This paper is a well-conducted systematic review looking at whether the cluster size of SW-CRTs vary and to what extent they vary. The paper looks at whether the trials used methods appropriate for unequal cluster sizes where necessary. This is a good paper, addressing a gap in the literature of SWCRTs.

My comments focus mainly on areas where the paper needs improvements in clarity or consistency in the reporting.

Response: Thank you for your kind comments, they have helped to improve the paper. Specific responses to each of your comments are given below.

Comment: Essential

The primary outcome stated in the abstract does not match up with the aims stated at the end of the background section, please change to reflect the main paper. The result in the abstract that 45% of trials with vary cluster size used an “appropriate” analysis is not reported in the main paper, please add this.

Response: The outcomes in the abstract have been changed to reflect the aims stated at the end of the background section. The result given in the abstract can be found in the discussion, but a sentence has now been added so that this result can also be found in the results.

Comment: At the bottom of the third paragraph of the background, the paper by Baio et al is given as a reference for an expectation that there will be a loss of power from unequal cluster sizes in SWCRTs. Baio did not look at the effect of unequal cluster sizes on power so this reference does not support this statement. Please either find a different reference or edit the statement to reflect that it is not supported by any literature.

Response: Baio et al state that “we would expect a loss of power if the cluster sizes vary substantially”, but we understand that the reference might be misunderstood and so the reference to Baio et al has now been removed.

Comment: It is unclear in the paper what is meant by an appropriate analysis with unequal cluster sizes, please clarify this. Some analysis methods are less efficient with unequal cluster sizes, but as far as I know they still give valid estimates and inference, so may be chosen despite their inefficiency. For example, in the fourth paragraph of the background, you should clarify that the methods you state are preferred when cluster sizes are unequal because they are more efficient.

Response: Thank you for bringing this to our attention. The criteria for determining when an appropriate method of analysis has been used have been clarified. We have also clarified that these methods are those that have been shown to be efficient when cluster sizes vary.

Comment: It is also unclear which of the analysis methods in the results are “appropriate” or “inappropriate” for unequal cluster sizes, and which were from trials with unequal cluster sizes. This makes any interpretation of the analysis methods results difficult. Please add some comments on which methods were used with equal vs unequal clusters, and which were “appropriate” for this.

Response: The number of studies that indicated that cluster were known to vary in size and used an appropriate method of analysis, and the number of studies that were shown to have varying cluster sizes and used an appropriate method of analysis, have been added to the results.

Comment: Please use a different measure of the spread of results throughout; the range isn't very informative. An interquartile range would be more informative.

Response: Ranges have been replaced with interquartile ranges throughout the paper.

Comment: In table one, the row name “rate” is on the same line as “continuous”. I think continuous is meant to be on the next line.

Response: This typo has been corrected so that “rate” and “continuous” are on separate lines.

Comment: What is the difference between a rate outcome and a time to event outcome? Similarly, what is the difference between a proportion outcome and a binary outcome? What do you mean by a discrete outcome? Please clarify.

Response: The outcomes have been reclassified in order to avoid any confusion. Rate and proportion have been grouped under binary and the discrete outcomes, on closer inspection were found to be ordinal and so renamed. Time-to-event outcomes are kept separate to rate outcomes as the analysis methods are very distinct.

Comment: On page 8 paragraph 1, please give a median or mean for the difference in the actual and required sample size, as well as a measure of the spread.

Response: The median and interquartile range for each measure of difference in actual and required cluster size have been added to the “comparison of cluster sizes” section of the results.

Comment: It would be informative to add details of how you estimated cluster size where the paper gave partial information.

Response: More detail has been added to the methods to explain how cluster sizes were estimated where the paper only gave partial information.

Comment: Optional

Since you report the proportion that accounted for unequal cluster size in the randomisation, you could add something in the background about why and when this should be considered.

Response: Thank you for your suggestion. A paragraph has been added to the background explaining the importance of accounting for unequal cluster sizes in the randomisation.

Comment: You could add whether studies accounted for varying cluster size in the randomisation to table one.

Response: Whether studies accounted for varying cluster size in the randomisation has been added to table 1.

Comment: Page 7, methods of analysis section: you could add that the simple analysis methods also fail to adjust for period effects as well as failing to account for clustering.

Response: The point that simple analysis methods also fail to adjust for period effects has been added to the methods of analysis section.

Comment: In the discussion, you highlight that the paper for continuous cross-sectional designs is limited to cross-sectional designs. You could also highlight that this only considered an ICC of 0.05; this is high for health research. Unequal cluster sizes may have a larger impact when the ICC is lower.

Response: A sentence has been added to the discussion to highlight that the referenced paper only considers a specific scenario and that with a different ICC the results may have been different.

VERSION 2 – REVIEW

REVIEWER	Mona Kanaan University of York United Kingdom
REVIEW RETURNED	30-Aug-2017

GENERAL COMMENTS	Thanks for responding to my earlier comments. Couple of minor points. Sentence spanning lines 108 and 109 (in the traced version); the text within the brackets needs clarifying or deleting. Please note that "clusters randomised to switch to the intervention at the same time" gives the impression that it is the norm to have multiple clusters switching to the intervention arm at a given randomisation point. Can Table 1 be presented in a better format? Line 278 (in the traced version) replace "accounted" with "account"
---

REVIEWER	Michael J. Grayling MRC Biostatistics Unit, Cambridge
REVIEW RETURNED	02-Aug-2017

GENERAL COMMENTS	Summary In my review of the previous version of this paper, I expressed several concerns regarding its content. The authors have now addressed these points well. Consequently, I have no further concerns regarding the manuscript. 1. Major Comments None. 2. Minor Comments None. 3. Typographical/Grammatical The authors may find the following suggested corrections sensible: 3.1. L266: Should be "account", not "accounted".
--

REVIEWER	Fan Li Department of Biostatistics and Bioinformatics No Competing Interest
REVIEW RETURNED	16-Aug-2017

GENERAL COMMENTS	No further comments.
----------------------

REVIEWER	Mike Campbell University of Sheffield UK I am on the CONSORT working group for reporting of stepped wedge trials
REVIEW RETURNED	08-Aug-2017

GENERAL COMMENTS	I can see I was a bit negative about this study in my first review, and the authors much improved it. I have only two minor grammatical corrections. Page 2 line 80 'first'-> 'former' Page 3 line 95 'normal data'=> 'continuous data plausibly Normally distributed'
---

REVIEWER	Jennifer A Thompson London School of Hygiene and Tropical Medicine, UK Both myself and the main author are part of a larger group developing CONSORT guidelines for stepped wedge trials.
REVIEW RETURNED	17-Aug-2017

GENERAL COMMENTS	The authors have responded to all of my comments and the paper is reading well. I have a couple outstanding query about the author's responses. Please can you clarify your comment "Rate and proportion have been grouped under binary... Time-to-event outcomes are kept separate to rate outcomes as the analysis methods are very distinct". My understanding is that rates and time-to-event outcomes are the same (the time to an event is 1/rate) usually analysed using a poisson, cox, or similar model. This is different to a binary outcome, where there is no known time scale in which the outcome occurs and a logistic model usually is used. Perhaps we have been using rate to mean different things, but it is important to get the outcomes grouped correctly in the paper. Page 10 paragraph one- you are arguing that the actual cluster size varied a lot from the required cluster size, but half of the trials used clusters within 10% of the required size. This seems good to me and warrants a mention in the discussion and in the abstract where you currently only give the range, although I agree with your comments about the outlying values being extreme.
---

VERSION 2 – AUTHOR RESPONSE

Reviewer: 1

Reviewer Name: Mona Kanaan

Institution and Country: University of York, United Kingdom

Please state any competing interests: None declared

Please leave your comments for the authors below

Thanks for responding to my earlier comments. Couple of minor points.

Comment: Sentence spanning lines 108 and 109 (in the traced version); the text within the brackets needs clarifying or deleting. Please note that "clusters randomised to switch to the intervention at the same time" gives the impression that it is the norm to have multiple clusters switching to the intervention arm at a given randomisation point.

Response: Thank you for your helpful comments. For clarification this sentence has been changed.

Comment: Can Table 1 be presented in a better format?

Response: Changes have been made to Table 1 to improve the format.

Comment: Line 278 (in the traced version) replace "accounted" with "account"

Response: Thank you. This typo has been corrected.

Reviewer: 2

Reviewer Name: Michael J. Grayling

Institution and Country: MRC Biostatistics Unit, Cambridge

Please state any competing interests: None declared

Please leave your comments for the authors below

Comment: Summary

In my review of the previous version of this paper, I expressed several concerns regarding its content. The authors have now addressed these points well. Consequently, I have no further concerns regarding the manuscript.

1. Major Comments

None.

2. Minor Comments

None.

3. Typographical/Grammatical

The authors may find the following suggested corrections sensible:

3.1. L266: Should be "account", not "accounted".

Response: Thank you for your kind comments. We have corrected this typo.

Reviewer: 3

Reviewer Name: Fan Li

Institution and Country: Department of Biostatistics and Bioinformatics

Please state any competing interests: None

Please leave your comments for the authors below

Thank you, we are happy we were able to address your previous comments.

Reviewer: 4

Reviewer Name: Mike Campbell

Institution and Country: University of Sheffield, UK

Please state any competing interests: I am on the CONSORT working group for reporting of stepped wedge trials

Please leave your comments for the authors below

Comment: I can see I was a bit negative about this study in my first review, and the authors much improved it.

I have only two minor grammatical corrections.

Page 2 line 80 'first'-> 'former'

Page 3 line 95 'normal data'=> 'continuous data plausibly Normally distributed'

Response: Thank you for your kind comments. We have corrected both of these typos.

Reviewer: 5

Reviewer Name: Jennifer A Thompson

Institution and Country: London School of Hygiene and Tropical Medicine, UK

Please state any competing interests: Both myself and the main author are part of a larger group developing CONSORT guidelines for stepped wedge trials.

Please leave your comments for the authors below

The authors have responded to all of my comments and the paper is reading well. I have a couple outstanding queries about the author's responses.

Comment: Please can you clarify your comment "Rate and proportion have been grouped under binary... Time-to-event outcomes are kept separate to rate outcomes as the analysis methods are very distinct".

My understanding is that rates and time-to-event outcomes are the same (the time to an event is $1/\text{rate}$) usually analysed using a poisson, cox, or similar model. This is different to a binary outcome, where there is no known time scale in which the outcome occurs and a logistic model usually is used. Perhaps we have been using rate to mean different things, but it is important to get the outcomes grouped correctly in the paper.

Response: Thank you for expanding on your previous comment. We agree with your understanding of rates and have now changed the grouping of the outcomes. Rate outcomes have now been combined with time-to-event outcomes.

Comment: Page 10 paragraph one- you are arguing that the actual cluster size varied a lot from the required cluster size, but half of the trials used clusters within 10% of the required size. This seems good to me and warrants a mention in the discussion and in the abstract where you currently only give the range, although I agree with your comments about the outlying values being extreme.

Response: Thank you for your comment. The 10% relates to those trials that presented the actual cluster sizes, this along with the result for those trials that presented summary measures of the actual cluster sizes (22%) have been added to the discussion. Unfortunately the word limit for the abstract is too restrictive to allow the addition of these results but we hope that what we have added to the discussion will be sufficient.

VERSION 3 – REVIEW

REVIEWER	Jennifer Thompdon London School of Hygiene and Tropical Medicine, UK Myself and the first author are part of the working group for stepped wedge trial CONSORT guidelines
REVIEW RETURNED	03-Oct-2017
GENERAL COMMENTS	All my comments have been addressed, and I have no further comments on this manuscript